# Determining the Characteristics of Faith-Themed Routes in Order to Receive an International Certificate: Studies on St. Paul's Travels

**Meryem Elif Çelebi Karakök**

Department of Architecture, Akdeniz University, Antalya 38000, Türkiye; elifcelebi@akdeniz.edu.tr

**Abstract:** The religious journeys of humanity and their components are now recognized as cultural heritage values. UNESCO, WHC, ICOMOS, CIIC, and COE are organizations that actively work and issue international certificates for the protection, promotion, and survival of religious routes. These organizations have certified 14 faith-based routes as of 2023. A route's certification is critical since it allows the route to be recognized globally and accessible to international tourism. However, each institution has its own set of requirements to obtain these certificates. When all religious cultures are examined, 14 routes are insufficient to explain the phenomenon of religion to today's people. For this reason, it is beneficial to increase the current number by re-activating the religious routes that have affected large masses. Many countries apply every year to obtain certificates from these organizations with various route studies. However, many applications are rejected as insufficient. Therefore, the goal of this study is to determine the effective criteria for religious routes to receive international certification. In this regard, St. Paul's Route stands out for its extensive geography spanning 12 countries and international potential. St. Paul is regarded as the most important figure in the spread of Christianity from Jerusalem to Anatolia and Europe. However, the St. Paul Routes being implemented do not meet the criteria of the any certificates. But the authentic St. Paul Route has the potential to receive certification from all organizations. In this research, the criteria required for the St. Paul Route to be certified by international organizations were investigated. A certified St. Paul Route will benefit many issues, including inter-religious dialogue between 12 countries, international cooperation, world peace, and sustainable tourism. It is thought to be an exemplary route with these features.

**Keywords:** faith-themed routes; St. Paul's route; cultural route certificates; cultural route certificate criteria

## 1. Introduction

Religion has always existed in human life, as evidenced by Göbeklitepe (Magli 2016) dated to 10,000 B.C., Gantija (Sparavigna 2016) dated to 5500 B.C., and Stonehenge (Pearson et al. 2007) dated to 2300 B.C. Mankind has written hymns for their religious beliefs, created musical instruments, danced, sewed clothes, performed rituals, sacrificed, fought to the death, built magnificent structures, and traveled long and difficult distances. All of humanity's religious activities and products are now recognized as important components of cultural heritage's tangible and intangible values. As a consequence, religious heritage values have to be protected and preserved (UNESCO 2003).

UNESCO, the World Heritage Committee (WHC), the International Council on Monuments and Sites (ICOMOS), and the International Committee on Cultural Routes (CIIC) are among the important organizations actively working on the protection, promotion, and survival of cultural heritage. They work on identifying, documenting, maintaining, and transferring cultural heritage elements to cultural routes, including religious heritage. The Council of Europe (COE), which has a similar mission, gives greater visibility to routes related to religious people and religious buildings. COE is the international organization

that organizes the most faith-themed routes; as of 2023, 12 of the 48 cultural routes are religious-themed (COE-CR 2023). UNESCO and WHC, as well as ICOMOS and CIIC, support route studies, nevertheless not as effectively as COE. Religious routes comprise four of the six cultural routes designated by UNESCO and the World Heritage Committee. Only two of these routes differ from the COE route list (UNESCO and WHC 2023). Only one of ICOMOS and CIIC's four cultural routes is religious, and it is also on the COE's list (ICOMOS and CIIC 2023). These organizations award certificates to routes that meet their criteria. A route's certification is critical since it allows the route to be recognized globally and makes it accessible to international tourism. In this case, the world has only 14 internationally certificated religious routes. These routes continue the historical cycle by introducing the tangible and intangible religious legacy into current society through sustainable tourism.

On the other hand, when all religious cultures worldwide are examined, it is considered that the 14 routes are insufficient to convey the phenomenon of religion to people today. Therefore, there is a need to increase the current number by re-activating the religious routes that have affected large masses. Many countries apply every year to obtain certificates from these organizations with various route studies. However, many applications are rejected as insufficient. Therefore, the goal of this study is to determine the effective criteria for religious routes to receive international certification. In this regard, the St. Paul's Travels Route, which Greece and Turkey work on individually, draws attention in the new religious route planning paradigm. St. Paul, who traveled to spread Christianity in the first century A.C., made four missionary trips. He is regarded as the most influential figure in the spread of Christianity from Jerusalem to Anatolia and then to Europe. Many churches in the Middle East, Turkey, and numerous European countries were constructed in his honor (IPDCT n.d.).

A cultural route is defined by the Council of Europe as the development of a travel route, a historical journey, a cultural concept, figure, or phenomenon of international importance for understanding Europe's common values and its transformation into a cultural, educational, heritage, and tourism cooperation project (COE-AR 2018). The UNESCO Cultural Heritage Committee accepted the concept of the cultural route in 1994, defining it as "the path/trace of the heritage consisting of concrete elements, where the cultural importance arising from the exchange and multidimensional dialogue between countries and regions is seen temporally and spatially along the route" (UNESCO and WHC 1994). ICOMOS defines the cultural route as an interactive and dynamic representation of different peoples' rich intercultural diversity and contributions to cultural heritage (ICOMOS 2008).

The transnational importance and richness of its historical, cultural, religious, geographical, and anthropological findings show that the St. Paul's Travels route has the criteria to be certified by these organizations. The route of St. Paul is a clear and detailed account of how Christianity spread from Asia to Europe. In order to fully comprehend the purpose, narrative, objectives, and implications of St. Paul's journeys, a reorganization to include all stops is necessary. To date, due to political and economic factors, the execution of St. Paul's travels has not been comprehensive and authentic. Greece's proposal includes Greece, Southern Cyprus, Italy, and Spain, where St. Paul may or may not have visited. According to belief, St. Paul visited Syria and many other parts of Turkey, but Syria cannot be visited today due to sociopolitical reasons. However, this does not explain why Turkey excludes St. Paul's stops from its scope (Baniotopoulou 2016). The St. Paul's Route in Turkey, on the other hand, runs for 500 km from Antalya's Perge Ancient City to Isparta's Antioch Ancient City. It is organized and implemented as a small local-route trip by a small Christian non-governmental organization in Antalya (CRS 2012). Consequently, neither route implementation has found St. Paul's route's true value and meaning.

The St. Paul's Route depicts the missionary journeys as narrated in the Bible. It holds religious, historical, and authenticity value. The four different travel routes it includes, as well as their actual stops, should be explained. All historical, geographical, religious, architectural, tangible, and intangible cultural data should be accurately transmitted to

current and future generations. The current practices are carried out in a deceptive manner. For example, a tourist who follows the route suggested by Greece will believe that St. Paul only traveled within the European continent, whereas a tourist who follows the route in Turkey will believe that St. Paul only walked 500 km in Turkey. This situation can cause alarming and funny situations in society.

However, because the majority of the European population is already Christian, the memories of St. Paul and the religious buildings built in his honor are preserved. The main issue is being addressed in Turkey and the Middle Eastern countries. In Turkey, for example, churches have been built in the name of St. Paul. Nevertheless, these churches are no longer in use because they lack a congregation. There are also church structures that were built recently and have historical and architectural value. The vast majority of these churches do not have a congregation. A significant portion of the churches are owned by Greece. Yet, Turkey and Greece face political challenges in restoring and utilizing these churches. Therefore, these churches had been abandoned and were on the verge of collapsing. As a result of St. Paul's efforts to spread his religion, these churches can be included in the cultural route, opened for use, re-functioned, and thus passed on to future generations. One of the goals of cultural routes is to protect all types of cultural heritage in the surrounding area. Another of the goals of cultural routes is to provide economic benefit and employment to the communities where they stop. The global economic crisis will be beneficial to the people at these stops because it will raise awareness about cultural heritage protection and develop their protective instinct. Since St. Paul visited 12 different countries on four separate trips, the inclusion of all these countries on the route will make St. Paul's Route richer, interesting, and instructive. More importantly, it will foster tolerance, mutual understanding, and feelings of peace among nations, peoples, and religions; it will fortify cultural ties between the European and Asian continents.

## 2. Literature Review

When the literature review was conducted, there were few publications about cultural routes. According to Vada et al. (2022), there are 43 peer-reviewed articles published on cultural routes, which is insufficient to develop new routes. Furthermore, the number of academic publications on routes has decreased even more in recent years, which they attribute to the COVID-19 pandemic. When religious routes are removed from the publication list, it is obvious that the number of publications will be much lower.

It has been determined that the literature examined within the scope of this research focuses on the definition of the religious route, the reasons for its preference, the features it possesses or lacks, the opportunities it provides, or the threats it faces. They also mentioned the issues that must be considered and implemented to increase the potential of pilgrimage routes, better protect and manage them, and correct deficiencies. Pilgrimage routes are an economic, social, and spatial phenomenon, according to McGrath (1999), who attempted to define the religious route. It is the geographical experience of going to a place of worship or a holy place. According to Mahanti (2022), a religion is held by approximately 93% of the world's population. At some point in a person's life, the desire to be one with God takes precedence and drives them to visit religious sites. The divine atmosphere in these locations combines with religious structures to create an appealing environment for visitors. Temples, churches, mosques, shrines, cathedrals, gurdwaras, and synagogues are magnificent religious structures that make you feel as if you are meeting with God. In various sources, words like "faith-based discoveries, pilgrimages, and missionary journeys" are used to describe religious routes. Cultural routes are used as the top title and pilgrimage routes as the subtitle by organizations that issue international certificates. Pilgrimages are the most popular journeys around the world and have been carried out for centuries. Every year, the number of people who prefer these routes grows. According to the World Tourism Organization, the number of pilgrims to India's Haridwar has increased by 94% in the last five years; 87% to the Vatican; and 78% to Mecca in Saudi Arabia. Every year, approximately 300 to 350 million pilgrims and tourists visit certain religious sites.

Some researchers investigated why people choose religious-themed routes. According to Abbate and Di Nuovo (2013), the first reason travelers take religious routes is to be pilgrims, with the second reason being to explore or socialize. According to Wang et al. (2016), religious routes appeal to believers for two reasons: mental relaxation and cultural enjoyment. Cruz-Ruiz et al. (2020) observed a significant increase in the development of themed tourist routes; they stated that routes that combine architecture, scenery, gastronomy, and festivals are preferred. Religious routes, according to Choe et al. (2015), are also experienced by those who do not practice that religion. These visitors prefer religious routes for a variety of reasons, including learning about different cultures and lifestyles, getting away from city noise, pondering, and getting away from daily responsibilities. According to Kruger and Saayman (2016), the reasons for joining religious routes are spiritual satisfaction, inner peace, inner healing, faith discovery, community unity, personal satisfaction, and religious requirements. The route divides its travelers into three different categories: adherents, explorers, and seekers. According to Gutic et al. (2010), religious routes can also be designed to visit only one or a few religious buildings. The main motivation for such routes is the building's historical, architectural, and religious significance. Many cathedrals in Europe are considered sacred by the authorities, and visiting these structures is similar to being a pilgrim.

The themes of publications about the characteristics of religious routes can be divided into three categories: those that propose contemporary touches to the routes, those that attempt to define the components of the authenticity criteria, and those that attempt to establish their relationship with all types of cultural heritage. Shakiry (2006) suggests that the discovery of places of religious importance and the construction of quality tourism facilities in these places lead to an increase in pilgrimage tourism, but first and foremost, the beautification of the regions surrounding the pilgrimage sites and all necessary modern services, particularly the road and transportation system, should be provided. Tourist motivation is one of the most important factors to consider in faith-themed routes, according to Baibakov (2019). Aside from moral considerations, it is necessary to improve the quality of modern service, implement the appropriate pricing policy, and carefully plan advertising and information promotions. As stated by Amaro et al. (2019), since individuals from various geographies and cultures participate in pilgrimage routes, certain holy places must be visited in order to become a pilgrim, and there are prayers that must be performed. Therefore, there should be mobile phone applications that inform and guide. Given the likelihood of internet outages in rural areas, these applications should include offline functionality and the ability to use GPS for local identification. Furthermore, this application should be able to serve in multiple language groups by predicting the presence of pilgrims from various geographies and cultures. Yet again, pilgrims should provide detailed information about public services, lodging, and transportation while traveling on the route so that they are not concerned about these issues. Pilgrims will be able to focus more on spiritual and religious aspects on pilgrimage routes that offer all of these services.

Authenticity is important for pilgrims to experience holy places correctly and to feel spiritual values, according to (Kim et al. 2020), who attempt to define the components of the authenticity criterion. Some researchers have also worked on the definition of authenticity in religious pathways, explaining it through various concepts or what its components could be. Belhassen et al. (2008), for example, coined the term "theoplacity" to describe a person's devotion to a holy place based on religious belief and the spiritual connection they feel in that place. The author defined authenticity as having three components: belief, activities, and places. Wright (1965) coined the term "geopiety" to express the deep respect and devotion that people have to a certain place or geography due to their religious belief. Tuan (1976) discussed the term geopiety through the concept of authenticity, and how people perceive and make sense of religious geographical areas and places. According to Olsen and Wilkinson (2016), religious routes should be regarded as a slow travel experience since speed and dominant cultural norms diminish the spiritual value of these routes. Spiritual worth is an expression of uniqueness. The Cultural Route Assessment Model

(CREM) was created by Božić and Tomić (2016). One of the most important findings from the surveys they conducted with travelers in Serbia for the Roman Emperors Route for this model is that some parts of this heritage route were not considered authentic by the participants. Another important piece of information is the expectation that historical people and events will be supported by an interesting story. As an outcome, they argue that the authenticity of the route is defined by a real or mythological story and the places that support the story. According to Santos (2002), the uniqueness of a religious route is hidden in the primitiveness, difficulty, and distress of travel and accommodation facilities. Pilgrimages should not be built with ultramodern and comfortable facilities. Travelers who believe they will be unable to complete the entire route should begin at a convenient stop. Caton and Santos (2007) also supported this idea and stated that travelers also want to experience difficulties and reach personal maturation.

Researchers studying cultural heritage values have concentrated on the routes' tangible and intangible cultural heritage components. Naramski and Szromek (2019) proposed that cultural routes should be merged with tourism and have a strong network structure to protect cultural heritage values. As stated by Singh and Kumar (2022), intangible cultural heritage values, along with the route, temples, and other traditional structures, are important determinants of religious routes. Religious beliefs provide the spiritual meaning of religious routes' intangible cultural heritage values. Researchers explain this discourse through the natural elements that comprise the theme of India's five sacred routes, the meanings they discover in the human body, and the concepts they encounter in human life. In this regard, sacred activities, acts of worship, and sacred symbols should all be considered. On the other hand, cultural landscape is an essential component of religious routes. In this context, sacred trees, sacred public areas, sacred water sources, and sacred statues should be carefully considered in route planning.

According to Mishra (2000), religious routes are regarded as a first-rate industry all over the world. Regional development is required for job creation and the re-establishment of cultural values. As noted by Gupta (2006), pilgrimage tourism has a significant impact on socioeconomic change. Not only pilgrims but also tourists who are curious and interested in various pursuits participate in religious routes. As a result, the number of tourists taking religious pilgrimages grows rapidly each year. With the increase in tourists brought by religious routes, new professions that employ locals are emerging. Route paths with natural and cultural richness of underdeveloped countries have the potential to attract tourists seeking new and authentic experiences, according to Briednhann and Wickens (2004) and Mutana and Mukwada (2020). The fact that these countries are on international trade routes represents a significant opportunity for economic development. In accordance with Vijayanand (2012), religious routes attract new investments to its geography. Tourist income not only revitalizes the locals, but also provides the necessary funds for the protection and management of their religious centers. Many monasteries and church buildings, according to Krogmann et al. (2021), can be restored thanks to the proceeds of a pilgrimage route launched in 1993 in Slovakia. They claim that pilgrimage routes provide opportunities to restore and use religious structures.

According to some studies on the effects of tourism on religious routes, these routes and their components are vulnerable to some threats. These threats include overcrowding at the sanctuaries, the construction of too much infrastructure and superstructures in the immediate vicinity, the disappearance of open spaces, public spaces, and urban spaces, the region's excessive increase in real estate prices, and the need for locals to leave due to high costs. These include environmental pollution, the loss of a mystical atmosphere, identity issues, and conflicts between locals and tourists. In recent years, outbreaks such as the pandemic have stopped visits, posing a threat in the opposite direction of the afore-mentioned factors. According to Orland and Bellafiore (1990), a socio-economic threat researcher, pilgrimage tourism areas in developing countries are subject to extraordinary economic pressures and changes. Because they are considered sacred by the locals, these areas may be the last open areas of the geography where they are located. However, the



pressure of tourism and economic inputs causes the construction volume in the sanctuaries and their immediate surroundings to be excessively increased, and these areas are occupied by extremely crowded masses. As stated by Vijayanand (2012), religious routes not only generate income for the geography to which they belong, but they also raise costs. Garbage collection, water disposal, lighting, and marketing and promotion all necessitate a substantial budget. Real estate prices in the region rise, competition for land use begins, and the poor local population is sometimes forced to relocate. Given the threats to spiritual values, Hung et al. (2017) propose that conflicts between commercialization and sanctity be resolved by developing a balanced model for religious routes. Material facts should not be allowed to undermine the route's religious values and philosophy. Many touristic businesses, such as hotels, restaurants, and shops, have sprouted up near popular temples, according to Shinde (2007). This lessens the pilgrimage route's sacred atmosphere. The arrangement of religious routes with crowded tourist groups, according to Santos (2002), causes these routes to lose their religious and spiritual character. According to the research of Raj and Morpeth (2007), the Council of Europe's designation of the Santiago Pilgrimage Route as the European Cultural Route accelerated the secularization process. The conversion of local religious rituals into festivals destroys the spirit of faith. Terzić and Dogramadjieva (2022) investigated the benefits and drawbacks of bringing together racially and religiously diverse nations on a path where there had previously been disagreements. Through surveys conducted with 627 people in five different countries, they attempted to determine how the arrangement of the Ottoman Heritage Route in the Balkan countries would be received by the local people. In total, 27% of those polled responded negatively, 36% were undecided, and 37% responded positively. Therefore, careful consideration of national identities and religions in shaping cultural routes is required, as is mental preparation of local people for this route, as well as a very good international management plan and cooperation. Some researchers have recently investigated the effects of the pandemic threat, which has halted religious route travel, as it has in every sector. According to Mróz (2021), pilgrimages dropped by 90% during the first six months of the pandemic. Fear of death and feelings of refuge in religion increased interest in pilgrimage routes in the following periods. People under the age of 60 preferred to become virtual pilgrims using AR technology, while those over the age of 60 preferred to make pilgrimages on foot and with individual participation. While the number of road travelers has decreased, the number of annual pilgrims has increased. Tsironis investigated the effects of the pandemic period on St. Paul's Route in Greece, concluding that route travels, which came to a halt during the pandemic, resumed their previous speed immediately after the pandemic (Tsironis 2022). As a consequence, he stated that religious tourism, which began in ancient times and continues to exist today despite the passage of time, has proven to be resilient in the face of historical events, political fluctuations, health hazards, and security risks.

## 3. Decision-Making Organizations and Their Criteria for Choosing International Cultural Routes

The COE (Council of Europe), the United Nations Educational, Scientific and Cultural Organization (UNESCO) and World Heritage Committee (WHC), and the International Council on Monuments and Sites (ICOMOS) and International Committee on Cultural Routes (CIIC) actively work on cultural routes. The cultural route topic was first mentioned by the European Union (EU) in 1964 and COE was assigned to work on this topic. COE has developed a cultural routes program, established an institute, created a website and digital map system, established a certificate system, and has now implemented 48 cultural routes through agreements with stakeholders such as tourism companies and management centers. In 1994, UNESCO and ICOMOS met and developed a program to evaluate cultural routes. UNESCO and WHC cultural routes were included in the "World Heritage List" after ICOMOS established the CIIC committee to study cultural routes and published a charter (Table 1).



**Table 1.** Cultural Works of International Organizations.

| Date | Event |
|---|---|
| **Cultural Routes Works of COE** | |
| 1964 | European Union: Expert Group start to discuss cultural routes. |
| 1984 | Invite member states for "Cultural Routes of the Council of Europe Programme" |
| 1990 | Establishment of "European Institute of Cultural Routes-EICR" |
| 1993 | First cultural route was announced as "The Santiago de Compostela Pilgrim Route" |
| 2002 | Creation of "A Common Heritage: Cultural Routes and Landscaping" web portal. |
| 2010 | The Enlarged Partial Agreement on cultural routes (EPA) established |
| 2012 | Published "The Colmar Declaration" |
| 2012 | Established "Crossroads of Europe" which has established cooperation with tourism companies, authorities, and other relevant stakeholders. |
| **Cultural Routes Works of UNESCO, WHC and ICOMOS and CIIC Partnership** | |
| 1994 | UNESCO and ICOMOS announce "Part of Our Cultural Heritage: Cultural Routes" |
| 1998 | ICOMOS establish the "The International Committee of Cultural Routes (CIIC)" |
| 2005 | UNESCO and WHC announce cultural routes as a part of World Heritage List |
| 2008 | ICOMOS publish "Charter on Cultural Routes" |

COE, UNESCO and WHC, ICOMOS and CIIC have conducted research to define the concept of cultural route and its criteria (Table 2). The purpose of the COE's European Cultural Routes Program (COE-CR 1987) is to demonstrate the common and living cultural heritage of Europe's various countries and cultures through travel in space and time. To be accepted into this program, a route must first obtain a "Council of Europe Cultural Routes Label" certificate. The COE has a number of criteria for obtaining this certificate (COE 2013). Cultural routes were designated as one of the four parts of the World Heritage List by UNESCO and WHC as a global heritage in continuous and numerous interactions with the environment, with strategic, symbolic, philosophical, dynamic, and evocative dimensions. It has an established criteria for accepting cultural routes into the WHL (UNESCO and WHC 1994). ICOMOS has defined cultural routes as humanity's common cultural heritage, which should be protected through collaborative efforts. It is stated that different cultural groups should serve purposes such as increasing interaction, protecting and preserving cultural heritage, and creating a social, economic, and physical environment that is sustainable. With its cultural routes charter, ICOMOS has also published its own cultural routes criteria (ICOMOS 2008).

**Table 2.** COE, UNESCO and WHC, ICOMOS and CIIC cultural route criteria.

**COE Criteria (COE 2013)**

- Being a theme that represents European values and common to at least three European countries
- Being the subject of transnational, multidisciplinary, scientific research
- Contributing to the interpretation of Europe's contemporary diversity, as well as enhancing European memory, history, and heritage
- Supporting cultural and educational exchanges for young people
- Developing innovative projects in the field of cultural tourism and sustainable cultural development
- Developing touristic products for different groups

**Table 2.** *Cont.*

| UNESCO and WHC Criteria (UNESCO and WHC 2021) |
|---|
| ●   Should meet the main criteria of the World Heritage List: |
| ○   Outstanding Universal Values: should be one of 10 items (UNESCO and WHC 2005) <br> ○   Authenticity: cultural values should be expressed correctly <br> ○   Integrity: All features should be presented as a whole <br> ○   Protection and management requirements: should be well protected and well managed |
| ●   Concepts should: |
| ○   have movement, dynamics and continuity <br> ○   have a value above its components and cultural elements <br> ○   emphasize exchange and dialogue between countries or regions <br> ○   be multidimensional in religious, commercial, administrative, or other aspects |
| ●   Identifier elements should be as follows: |
| ○   Spatial characteristics: the length and diversity of a route should reflect the complexity of the links it maintains or sustains <br> ○   Temporal characteristics: must have sufficient time of existence for historical identity <br> ○   Cultural characteristics: should include cross-cultural aspects or influences, such as linking distant ethnic groups and their mutual advancement through exchange <br> ○   Role or purpose: should reflect how communities' values are used and contributed <br> ○   Legendary stories: should reflect mythological and symbolic values as well as realities |
| ●   A route must be properly defined, along with the important inheritance components that depend on it: |
| ○   Route boundaries: should have properly defined spatial, temporal and cultural boundaries <br> ○   Concentration points: start, stop, transfer, and end points should be known |
| **ICOMOS and CIIC Criteria (ICOMOS 2008)** |
| ●   Every cultural route should have uniqueness value in terms of the natural and built environment <br> ●   The temporal duration and historical significance of the various sections of the route as a whole should be considered <br> ●   It should reflect the multifaceted, ongoing, and mutual exchange of goods, ideas, information, and values between people, continents, countries, and regions. <br> ●   It should protect the affected cultures' tangible and intangible heritage values <br> ●   It should engage with the natural environment <br> ●   It should integrate historical relations with cultural characteristics <br> ●   It should promote interaction between people from various cultures or ethnic groups <br> ●   It should have cultural characteristics rooted in the traditional life of different communities <br> ●   It should have cultural practices such as common heritage elements of different communities, ritual and celebration, etc. |

COE, UNESCO and WHC, ICOMOS and CIIC issued certificates to routes based on their criteria (Table 3) and published them on their websites. The cultural routes have been organized by theme by the COE. UNESCO and WHC have included their routes in the World Heritage List (WHL), which can be understood from the terms route and road. ICOMOS and CIIC routes are published on the CIIC web page with brief information and

maps. It redirects to the WHL page for detailed information. Religious routes are definitely included in the lists of all organizations. Table 3 shows religious routes in red, while other routes are in black. It is worth noting that the Santiago de Compostela Pilgrim Routes, the first approved route by COE, UNESCO, and WHC, is a pilgrim route that has received certificates from all three organizations.

**Table 3.** COE, UNESCO and WHC, ICOMOS and CIIC cultural routes.

| COE Routes (COE-CR 2023) | | |
|---|---|---|
| **Date** | **Cultural Routes** | |
| 1987 | Santiago De Compostela Pilgrim Routes | Religious Route |
| 1991 | The Hansa | |
| 1993 | Viking Route | |
| 1994 | Via Francigena | Religious Route |
| 1997 | Routes of El Legado Andalusí | Religious Route |
| 2003 | Phoenicians' Route, Iron Route in the Pyrenees | |
| 2004 | European Mozart Ways | |
| 2004 | European Route of Jewish Heritage | Religious Route |
| 2005 | Saint Martin of Tours Route | Religious Route |
| 2007 | Transromanica—The Romanesque Routes of European Heritage | |
| 2009 | Iter Vitis Route | |
| 2010 | European Route of Cistercian Abbeys | Religious Route |
| 2010 | European Cemeteries Route | Religious Route |
| 2010 | Prehistoric Rock Art Trails, European Route of Historic Thermal Towns | |
| 2010 | Route of Saint Olav Ways | Religious Route |
| 2012 | European Route of Ceramics | |
| 2013 | European Route of Megalithic Culture, Huguenot and Waldensian Trail | |
| 2014 | Atrium—Architecture of Totalitarian Regimes of the 20th Century In Europe's Urban Memory | |
| 2014 | Réseau Art Nouveau Network, Via Habsburg | |
| 2015 | Roman Emperors and Danube Wine Route, European Routes of Emperor Charles V, Destination Napoleon, In the Footsteps of Robert Louis Stevenson | |
| 2016 | Fortified Towns of the Grande Region | |
| 2018 | Impressionisms Routes, Via Charlemagne | |
| 2019 | European Route of Industrial Heritage, Iron Curtain Trail, Le Corbusier Destinations, Liberation Route Europe | |
| 2019 | Routes of Reformation | Religious Route |
| 2020 | European Route of Historic Gardens | |
| 2020 | Via Romea Germanica | Religious Route |
| 2021 | Aeneas Route, Alvar Aalto Route—20th Century Architecture and Design | |
| 2021 | Cyril and Methodius Route | Religious Route |
| 2021 | European Route d'Artagnan, Iron Age Danube Route | |
| 2022 | Historic Cafés Route, European Fairy Tale Route, Women Writers Route | |
| **UNESCO and WHC Cultural Routes (UNESCO and WHC 2023)** | | |
| 1993 | Routes of Santiago de Compostela: Camino Francés and Routes of Northern Spain | Religious Route |
| 1998 | Routes of Santiago de Compostela in France | Religious Route |
| 2004 | Sacred Sites and Pilgrimage Routes in the Kii Mountain Range | Religious Route |
| 2005 | Incense Route—Desert Cities in the Negev | |
| 2012 | Birthplace of Jesus: Church of the Nativity and the Pilgrimage Route, Bethlehem | Religious Route |
| 2014 | Silk Roads: the Routes Network of Chang'an–Tianshan Corridor | |
| 2014 | Qhapaq Ñan, Andean Road System | |
| **ICOMOS and CIIC Cultural Routes (ICOMOS and CIIC 2023)** | | |
| 2005 | Route of the Incense in the Negev | |
| 2010 | Camino Real de Tierra Adentro | |
| 2014 | The Silk Road | |
| 2014 | Qhapaq Ñan or Path of the Lord | |
| 2015 | Caminos de Santiago de Compostela: French Camino and Caminos del Norte de Spain | Religious Route |

## 4. Analysis of Certified Pilgrimage Routes

The COE, UNESCO and WHC, ICOMOS and CIIC organizations define the cultural routes for which they award certificates on their respective websites. Under various headings, these pages explain how the routes meet the certification requirements.

### 4.1. COE-Certified Religious Routes

The most effective and most certified COE at the international level has certified 12 religious routes. Each route has its own page on the COE's website. The criteria of the COE are explained on these pages under the headings of "theme, traveling today, heritage and council of Europe values". The theme section contains detailed information about the route's historical background, the traveling today section contains information about the partner countries, the length of the route, the mode of travel, and its widespread impact, the heritage section contains information about the route's tangible and intangible cultural heritages, and the council of Europe values section contains information about the route's significance for the European Continent (COE-CR 2023).

The theme section describes the route's history, heroes, story, and journey. The route's start and end points are specified (COE-CR 2023). The theme of the Santiago de Compostela Pilgrim Routes arose from a legend. According to legend, the body of St. James, which was brought to Spain from Jerusalem in 40 A.D. and forgotten, was discovered in the Cathedral of Santiago in the 9th century, and St. James was declared the protector of Spain. Beginning on this date, Christians began to travel to St. James' tomb (COE-SCPR 1987). The theme of Via Francigena was born from a diary. Sigeric, archbishop of Canterbury in England, was named Pope XV in A.D. 990. He traveled to Rome in order to purchase an investiture pallium from John. It records the journey's 79 stages. A pilgrimage route between England and Italy is created based on this diary (COE-VF 1994). The theme of the Routes of El Legado Andalus was inspired by religious history. The Andalusian Umayyad State, founded between the 8th and 15th centuries by individuals from the Arabian Peninsula, ruled the Iberian Peninsula of Europe. The journeys of these people, who brought Islam's faith and heritage to the European continent, have become a religious route today (COE-RELA 1997). The European Route of Jewish Heritage's theme is Jewish migration routes and heritage values. The migration movements of Jews from the Middle East to Europe, the Ottoman Empire, and America over an 18th-century span have been transformed into a religious route (COE-ERJH 2004).

All modes of transportation are available in the Traveling Today section. However, in order to feel the spiritual values on religious routes, some of the journeys are made on foot. Travelers on the Santiago de Compostela Pilgrim Routes travel by foot, bike, or horseback (COE-SCPR 1987). On foot, the Via Francigena Route rediscovers the land, history, and people (COE-VF 1994). Cruises in the Mediterranean are added to the land trips on the El Legado Andalus Routes (COE-RELA 1997). The European Route of Jewish Heritage has no restrictions on modes of transportation (COE-ERJH 2004). It is also worth noting that the routes have no distance restrictions. The longest distance on the Via Roma Germanica is about 2600 km (COE-VRG 2020), while the European Route of Jewish Heritage is 5555 km (COE-ERJH 2004). As a consequence, stops on the routes have been established, and travelers can join the route at any time. It has been noted that the routes include at least three European countries, but the total number of countries has not been determined. For example, the Via Romea Germanica has only three countries (COE-VRG 2020), and the European Route of Jewish Heritage has twenty-one (COE-ERJH 2004). Another important point is that after three European countries participate, countries from other continents can join the route as long as they are related to the theme. In El Legado Andalus Routes, three European countries joined Egypt, Lebanon, and Jordan (COE-RELA 1997). Turkey and Azerbaijan have also joined the European Route of Jewish Heritage's 19 European countries (COE-ERJH 2004).

The Heritage section introduces travelers to the route's tangible and intangible cultural heritage. For the Santiago de Compostela Pilgrim Routes, places of worship are added to

tangible heritage such as hospitals, lodging facilities, and bridges. Myths, legends, and songs are examples of intangible cultural heritage (COE-SCPR 1987). Traveling along the Via Francigena allows you to experience art cultures like Romanesque, Gothic, Renaissance, and Baroque. It provides a variety of values due to its various roles as a military, commercial, and pilgrimage route. It typically runs along major thoroughfares and is surrounded by historical monuments and archaeological sites (COE-VF 1994). El Legado Routes Andalus brings to life the Andalusian Umayyad State's impressive architectural heritage, literature, art, science, gastronomy, and traditions. It creates a bond between different ethnic groups (COE-RELA 1997). The European Route of Jewish Heritage includes Jewish heritage archaeological sites, synagogues, cemeteries, neighborhoods, and memories. The route also includes archives, libraries, and museums dedicated to Jewish history (COE-ERJH 2004).

The importance of the route for the European continent is stated in the Council of Europe Values section. As an instance, the Santiago de Compostela Pilgrim Routes are both a symbol of Europe's religious history and a model of cultural cooperation (COE-SCPR 1987). The Via Francigena is a mode of communication that contributes to Europe's cultural unity. It connects the cultures of Anglo-Saxon Europe and Latin Europe (COE-VRG 2020). El Legado Andalus Routes describes the meeting of Muslim and Christian cultures (COE-RELA 1997). The European Route of Jewish Heritage enriches Jewish culture and contributes to Europe's cultural diversity (COE-ERJH 2004).

### 4.2. UNESCO and WHC-Certified Religious Routes

UNESCO and WHC have certified four religious routes. Each route has its own page on the UNESCO and WHC website. On these pages, there are detailed descriptions of the routes under the headings "Description, Outstanding Universal Value and Criterias, Integrity, Authenticity, Protection and Management Requirements" (UNESCO and WHC 2023).

The religious identity of the route, the destination point, the countries it covers, the length, the extent of the area, the date of the first trip, the number of years it has been used, the date of inclusion on the World Heritage List, the holy places, the architectural heritage on the route, the archaeological sites, natural areas, and landscapes are all explained in the Description section (UNESCO and WHC 2023). Routes of Santiago de Compostela: Camino Francés and Routes of Northern Spain is an extension of a network of four Christian pilgrimage routes in Northern Spain. It is approximately 1500 km long. It has a total area of 16,285.7156 hectares. The route is thought to have been used since the tomb was discovered in the 9th century. In 1993, it was inscribed on the World Heritage List. It has a rich architectural heritage that includes cathedrals, churches, hospitals, hostels, and bridges (UNESCO and WHC-SCFNS 1993).

The Outstanding Universal Value section comprises Brief Synthesis and Criteria subsections. The description section is expanded upon in Brief Synthesis. Furthermore, heritage values considered to have outstanding universal value along the route are mentioned. These values are typically associated with areas of historical, religious, art, architecture, cultural landscape, and natural heritage. For example, Birthplace of Jesus: Church of the Nativity and the Pilgrimage Route, Bethlehem's outstanding universal value is that it is the birthplace of Jesus. Both Christians and Muslims revere the churches in Bethlehem, the route's destination (UNESCO and WHC-BPJ 2012). Sacred Sites and Pilgrimage Routes in the Kii Mountain Range have extraordinary universal value because they reflect a 1200-year-old sacred mountain tradition. This tradition reflects the interaction of the religions of Shinto and Buddhism (UNESCO and WHC-SSPR 2004). Rituals, festivals, beliefs, legends, and traditions associated with the route are also important components of the route. For example, the Sacred Sites and Pilgrimage Routes in the Kii Mountain Range's sacred mountain tradition, Birthplace of Jesus: Church of the Nativity and the Pilgrimage Route, and Bethlehem's Christmas celebrations are considered in this context. Furthermore, the dissemination of route information, people's contributions to the socioeconomic situation, and the establishment of cultural dialogue between travelers and locals are emphasized. Routes of Santiago de Compostela: Camino Francés and Routes of Northern Spain, for

example, have fostered a cultural dialogue between travelers and locals, thereby supporting local economic and social development (UNESCO and WHC-SCFNS 1993).

The Outstanding Universal Values criteria section has ten items, and cultural routes must meet at least one of these ten. Item vi, however, is not accepted on its own; it is evaluated in conjunction with one of the other items. The Criteria section explains briefly which of the ten criteria defines the Outstanding Universal Value feature. Some of the items, ii, iii, iv, and vi, can be found on the religious routes. Criteria ii determinants are factors that facilitate the exchange of religious values, such as interreligious interaction, being the birthplace of a religious belief, a religion contributing to the development of a settlement, and encouraging other communities to come here (UNESCO and WHC 2005). Routes of Santiago de Compostela: Camino Francés and Routes of Northern Spain play an important role in the cultural exchange between the Iberian Peninsula and the European continent (UNESCO and WHC-SCFNS 1993). The Kii Mountain Range Sacred Sites and Pilgrimage Routes is a synthesis of the East Asian religions of Shintoism and Buddhism (UNESCO and WHC-SSPR 2004). The Criterion (iii) determinants are that they are the only descriptors of a religion with its temples and rituals, or that they have witnessed the evolution of that religion over time. Sacred Sites and Pilgrimage Routes in the Kii Mountain Range bear witness to the evolution of Japan's religious culture over a thousand years (UNESCO and WHC-SSPR 2004). Criteria iv determinants are religious structures and religious cultural landscapes that are effective in religion development. Because religious development takes centuries, religious buildings and sanctuaries in archaeological and settlement areas are also considered determinants of this criterion. The Camino Francés and Routes of Northern Spain's historic buildings and residential areas (UNESCO and WHC-SCFNS 1993) and the Pilgrimage Routes in the Kii Mountain Range, the Kii Mountains are the 1200-year-old site of Japan's shrines and temples (UNESCO and WHC-SSPR 2004). Criteria vi is determined by its connection to a religious event, tradition, or belief of exceptional universal significance. This criterion is fulfilled by the Routes of Santiago de Compostela: Camino Francés and Routes of Northern Spain (UNESCO and WHC-SCFNS 1993), the power and influence of faith, the Sacred Sites and Pilgrimage Routes in the Kii Mountain Range (UNESCO and WHC-SSPR 2004), the sacred mountain tradition, the Birthplace of Jesus: Church of the Nativity and the Pilgrimage Route, the birth of Jesus in Bethlehem (UNESCO and WHC-BPJ 2012).

The components of religious routes are defined in the Integrity section, and it is emphasized that these components should be considered as a whole. Routes, settlements, access roads, buildings, sanctuaries, sacred places, lands, landscapes, archaeological sites, affected communities, contexts, processes, mnemonic forces and symbols, traditions, beliefs, ceremonies, crafts, and cultural exchanges are all components of religious routes. In route planning, all of these elements should be considered together. There should also be a buffer zone to protect the route. For example, temples and shrines in the Kii Mountain Range, paths, Shintoism–Buddhism–Shugen Sect beliefs and interactions, cultural landscapes, and a protective buffer zone all demonstrate integrity (UNESCO and WHC-SSPR 2004). Birthplace of Jesus: Church of the Nativity and the Pilgrimage Route includes the cave believed to be the birthplace of Jesus Christ in Bethlehem, the monastic community, the terraced lands, the ancient city and evidence of burials dating back to 2000 B.C., the main streets leading to the Church of the Nativity, religious ceremonies, traditions, workshops opening onto the streets and the protective buffer zone shows integrity (UNESCO and WHC-BPJ 2012). These buffer zones act as a protective barrier against possible dangers to preserve the values of the heritage site. Natural disasters, human activities, and industrial developments are all potential dangers. The most anticipated dangers are excessive tourism, dense construction, a rapid increase in the number of motor vehicles, insufficient parking space, environmental pollution, damage to historical buildings, and a lack of restoration.

The authenticity of the routes is determined by three different aspects in the Authenticity section: form and design, materials and substances, and use and function. The form and design of religious routes are determined by their characteristics of being a historical and real route. It should be based on a real person or event, retain historical details, and

be correctly expressed. Routes of Santiago de Compostela: Camino Francés and Routes of Northern Spain, for an instance, derive their authenticity value from the fact that their history and integrity have been preserved (UNESCO and WHC-SCFNS 1993). The primary materials and substances of religious routes are their routes, religious structures, rituals and practices. For example, the routes used in Routes of Santiago de Compostela in France (UNESCO and WHC-RSCF 1998); churches, temples, hospitals, bridges, rest stops; religious scenes and legends testify to their authenticity value. In Sacred Sites and Pilgrimage Routes in the Kii Mountain Range, the journeys and the tradition of building wooden structures are religious rituals and practices (UNESCO and WHC-SSPR 2004). The use and function of religious routes is related to the fact that the route and its components are still in use today. All religious routes of UNESCO and WHC are used by travelers and pilgrims today as they come from history.

The section on protection and management requirements includes the creation of protective laws and management plans. In this context, studies such as route and historical building registration, property determination, regulation of new construction conditions, natural site protection, authority determination, financial funds, maintenance periods for building and road maintenance and repair, and the creation of protective buffer zones should be carried out. Routes of Santiago de Compostela: Camino Francés and Routes of Northern Spain have been registered in the category of Historic Complex as the highest level of Cultural Interest (Bien de Interés Cultural) under the Spanish Historic Heritage Law. Crown Property owns the route, which is managed by the Jacobean Council (Consejo Jacobeo). This committee is in charge of furthering the route's promotion and cultural dissemination, preserving and restoring its historical and artistic heritage, organizing and promoting tourism, and assisting pilgrims. It is the responsibility of the municipalities through which the route passes to carry out activities such as industrial and urban growth and development, new transportation infrastructure like highways and railways, increased tourism, and pressure caused by the number of pilgrims, whilst not causing damage to the route (UNESCO and WHC-SCFNS 1993). The key principles and methodology for the conservation and management of Sacred Sites and Pilgrimage Routes in the Kii Mountain Range are outlined in the 2003 Comprehensive Conservation and Management Plan. The buildings along the route have been designated as National Treasures and Important Cultural Properties by the Japanese Law on Cultural Property Protection. The same law designates temple and tomb areas, pilgrimage routes, and the forest landscape surrounding them as Historical Sites, Places of Natural Beauty, and Natural Monuments. Relevant religious organizations are in charge of temple and shrine protection and maintenance. The national government finances and provides technical assistance for restoration and repair projects. The academic committees report on the route's protection and management status on a regular basis (UNESCO and WHC-SSPR 2004).

*4.3. ICOMOS and CIIC-Certified Religious Routes*

The only ICOMOS and CIIC-Certified religious route is Routes of Santiago de Compostela: Camino Francés and Routes of Northern Spain. The ICOMOS and CIIC page of this route leads to the UNESCO and WHC page. Therefore, the route has no definition for ICOMOS and CIIC (ICOMOS and CIIC 2023).

## 5. Materials and Methods

The authentic St. Paul's Route and the current St. Paul's Routes were discussed as material in this research. Through literature reviews, the descriptive analysis method was used as a method.

*5.1. Materials*

5.1.1. St. Paul's Authentic Journeys

St. Paul was a Roman citizen, Jewish Christian missionary, and the founder of the Pavlik churches who lived from 5 to 67 A.D. The Bible refers to him as Paul. He is widely

regarded as the most influential figure in the spread of Christianity from Jerusalem to Anatolia and then to Europe. The Acts of the Apostles section of the New Testament in the Bible contains information on St. Paul's life and travels. St. Paul, according to the New Testament, made four different journeys (Figure 1) for missionary purposes over a period of 20 years (Baniotopoulou 2016; CRS 2012; SPMJ n.d.).

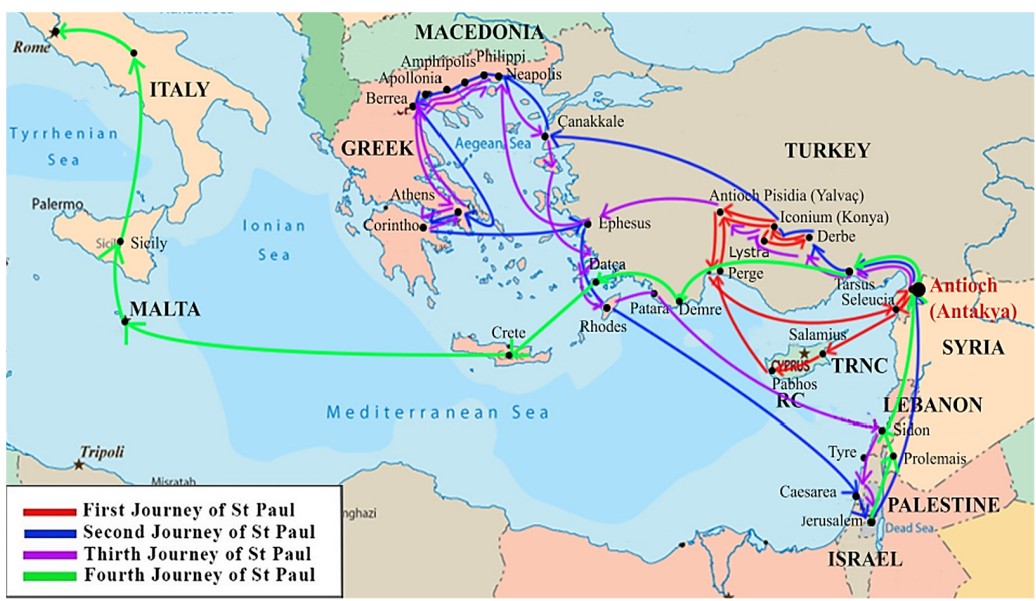

**Figure 1.** St. Paul journeys.

He made his first journey between the years 46 and 48 AD. He first travels by sea to Antioch (Antakya), Selecuia (Samandağ), Cyprus, and Attalia (Antalya) Marine, and then by road to Perge, Psidia Antioch (Yalvaç), and Iconium (Konya). He completes his first journey by returning to Antioch (Antakya) via the same route, without stopping in Cyprus (Baniotopoulou 2016; CRS 2012; SPMJ n.d.).

He made his Second Journey between 49 and 52 A.D. He traveled by road from Jerusalem to Antioch (Antakya), Tarsus, Iconium (Konya), Psidia Antioch (Yalvaç), and Troy (Çanakkale). He reached North Macedonia–Greece–Anatolia (Ephesus)–Rhodes Island–Syria–Jerusalem by sea from Troy (Çanakkale). Again, by road, he passed to Galatia–Phrygia–Antioch (Antakya) (Baniotopoulou 2016; CRS 2012; SPMJ n.d.).

He made his third journey between the years 53–57 A.D. He first traveled by road to Antioch (Antakya), Tarsus, Iconium (Konya), Psidia Antioch (Yalvaç), and Troy (Çanakkale). He traveled by sea to North Macedonia. He visited the cities of North Macedonia and Greece. He returned via Greece, Troy (Çanakkale), Kos Island, Rhodes Island, Kalkan, Tyre and Jerusalem (Baniotopoulou 2016; CRS 2012; SPMJ n.d.).

He made his fourth journey between 59–69 A.D. However, this is a mandatory journey. In Jerusalem, St. Paul was arrested and judged by Roman authorities. Following the trial, he was transported to Rome via Syria–Sidon–Antioch (Antakya–Tarsus–Demre–Datça–Crete Island–Malta Island–Sicily–Italy). St. Paul was imprisoned here before being executed in 64 or 67 A.D. (Baniotopoulou 2016; CRS 2012; SPMJ n.d.).

5.1.2. Currently Used St. Paul Routes

There are many St. Paul's Routes practiced today. However, one route led by Greece and another route implemented in Turkey are quite active and popular. In addition, both practices have efforts to obtain certification from international organizations. Therefore, in this study, these two routes were chosen as the current St. Paul Route applications.

St. Paul's Route of Greece

The European Grouping of Territorial Cooperation (EGTC) was formed by the tourism networks of Greece, Italy, Northern Cyprus, and Belgium. They implemented the cultural route "In the footsteps of St. Paul, the Apostle of the Nations" as a result of the Cult-RInG Interreg Europe project—Cultural Routes as Investments for Growth and Jobs, 2017–2021. They conducted numerous research and documentation studies on St. Paul's heritage in order to determine the route's content, the regions to be included in the route, the important places and touristic points. In addition, they attended various meetings, workshops, and discussions, as well as visited significant locations (EGTC 2021). Since 2006, the route has been in operation. It follows a route that includes stops in Southern Cyprus, Greece, Malta, Italy, and Spain (Figure 2). They hope to obtain a certificate for the route from the Council of Europe. They attempted to obtain a certificate from the COE with a route titled "In the Footsteps of St. Paul, the Apostle of the Nations-Cultural Route" during the 2021–2022 application period. It still does not have a certificate (Baniotopoulou 2016; CRS 2012; SPMJ n.d.).

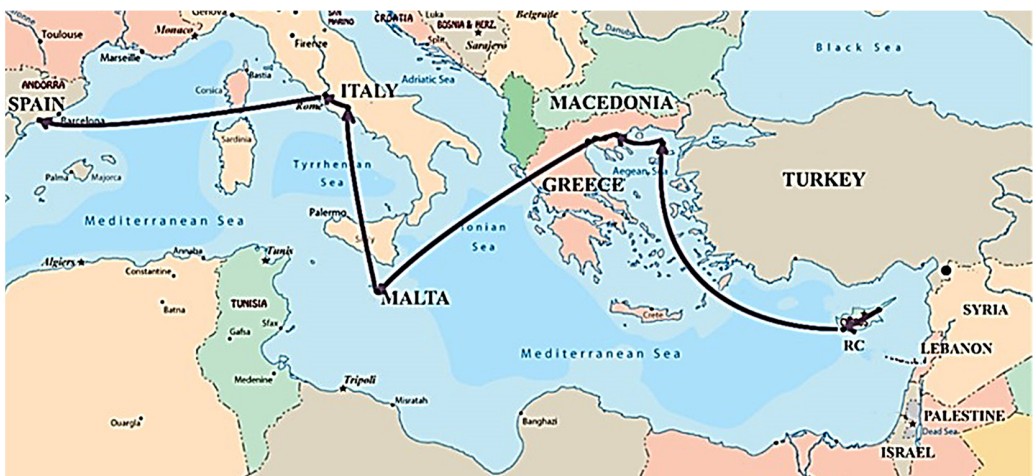

**Figure 2.** St. Paul's Route of Greece.

St. Paul's Route of Turkey

The St. Paul's Route in Turkey is a 500 km long path that takes 27 days to walk between the Perge Ancient City, which is located 10 km east of Attalia (Antalya), one of Turkey's western Mediterranean coastal cities, and the Psidia Antiochia Ancient City, which is located in the Yalvaç District of the mountainous city of Isparta (Figure 3). A second branch departs from Beşkonak at the entrance to Köprülü Canyon National Park, 80 km northeast of Antalya. In the Roman ancient city of Adada, the second branch connects with the first one. The route follows Roman roads, trails, and forest roads, and in places is suitable for mountain biking. Kate Clow established this route in 2008 to bring tourism to the countryside and to give hikers an insight into the countryside by passing through the locations where St. Paul went on his first trip to Asia Minor. Accommodation is available in village houses or small pensions along the road, but camping is done in long stages in designated camping areas (CRS 2012).

*5.2. Method*

The goal of this study is to identify the determining criteria for religious routes to receive international certification. The St. Paul's Route, which has a lot of potential, was chosen as the research material. As research methods, in-depth literature analysis and descriptive analysis methods were used. Firstly, it was determined what types of studies on religious routes had been conducted in the literature. According to the literature review, there are book publications about cultural routes that are specific to very old dates. The concepts associated with the cultural route are generally attempted to be explained in

the books. When scientific article research was examined, it was discovered that there are generally publications on cultural routes and few publications on religious routes. The articles were mostly about tourism, the socioeconomic impacts of cultural routes, the opportunities they provide, and the risks they face. There are, however, no publications in the literature on route certification systems, certification criteria, or how to present a route. A descriptive analysis of the certification that a route can receive has been made using information gathered from the literature and certificate programs.

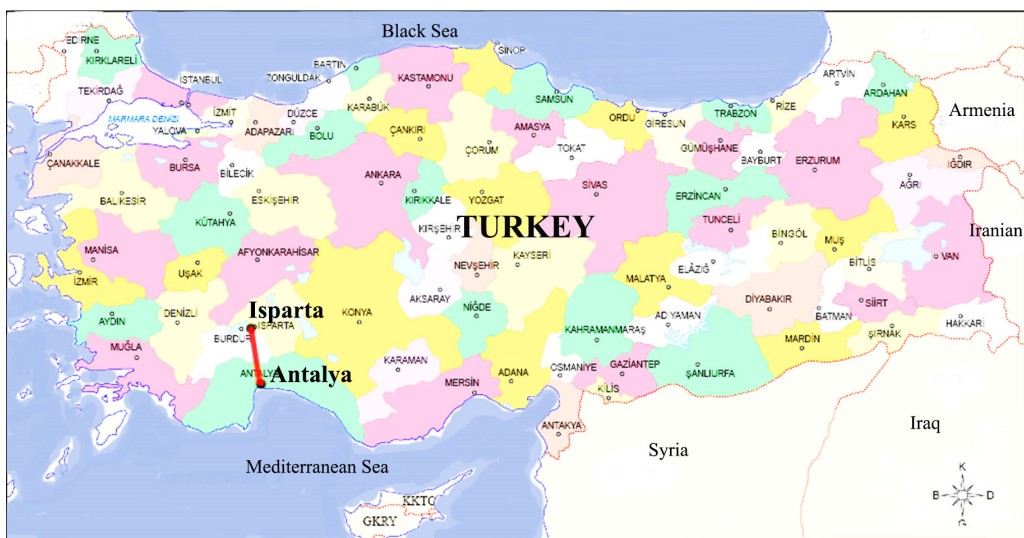

**Figure 3.** St. Paul's Route of Turkey.

This method's steps are indicated as "Determining decision makers for certification, determining of certificate criteria, researching of authentic route, researching authentic route, researching current versions of the authentic route, planning of the proposed route and eligibility analysis of the proposed route for certificate" (Figure 4). This method will determine how a route should be planned as well as which certificate programs the route can apply to.

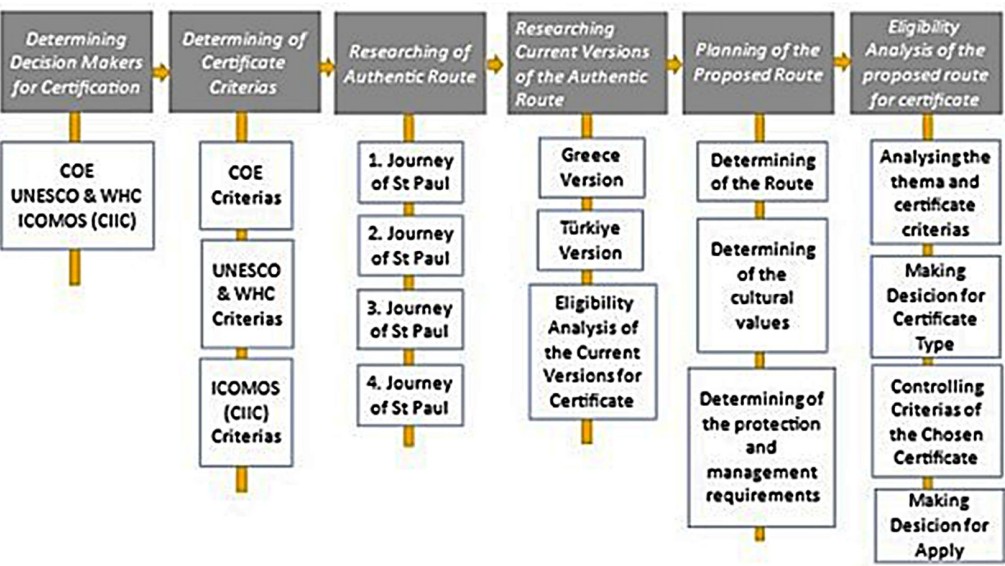

**Figure 4.** Flowchart of the research.

### 6. Conclusions and Recomendations

Cultural routes are one of the most important and current issues at the intersection of architectural conservation and tourism disciplines today. The goal of organizing a cultural route is to convey important information and values about historical roads while connecting them and presenting travel, recreation, observation, sports, entertainment, and/or discovery routes as a whole. Thus, cultural routes (CR) promote cultural heritage as humanity's common heritage and encourage the spread of cultural activities. They aid in the discovery and protection of cultural values such as historical cities and villages, architectural heritage, cultural landscape, and intangible cultural heritage, and they are also useful for evaluating natural areas in this context. They expand economic and social development opportunities, particularly in terms of job creation, by bringing movement and dynamism to the regions within its sphere of influence. The potential benefits and advantages drive cultural tourism and influence the development of sustainable tourism.

The St. Paul's Route was used to evaluate the characteristics that a religious route should have based on the results of an analysis of the criteria of the international organizations that grant certificates and the characteristics of the certified religious routes. The versions of the St. Paul's Route in use today are not of sufficient quality to receive international certification. However, when compared to the authentic St. Paul's journeys, the St. Paul's Route has the potential to meet the needs of all three decision-making organizations. For this reason, in this study, a new St. Paul Route that can receive certificates from international organizations has been suggested, based on the real stops in St. Paul's authentic journeys (Figure 5).

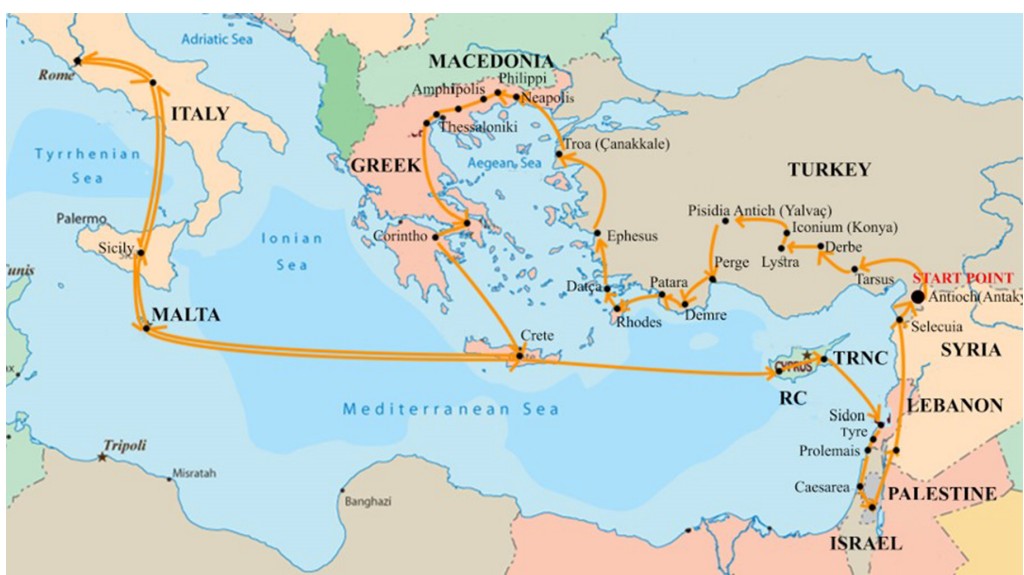

**Figure 5.** Suggested St. Paul's Route and Stops.

The authentic St. Paul's Route provides stops in 11 countries: Turkey, Greece, North Macedonia, Italy, Malta, Cyprus (northern and southern parts), Lebanon, Israel, Palestine, and Syria. It has stops in four European countries, as requested by the COE criteria: Greece, Italy, North Macedonia and Malta. All of the countries St. Paul visited were turned into a ring in this study. The longest distance on the route is 3530 km by road and 2305 km by air distance between Rome and Jerusalem. The longest distance traveled on COE pilgrimage routes is approximately 3000 km. Due to the length of the route, the St. Paul route should be designed in such a way that travelers can join the route from many stops, just like the Santiago de Compostela Pilgrim Route.

As the most influential person in the spread of the Christian religion to Europe, St. Paul has a special theme that represents European values. Scientific studies on St. Paul have been and continue to be conducted primarily in Greece and Turkey. Archaeological

excavations also aid religious and historical studies. It welcomes the expansion of Europe's religious memory, history, and heritage. Many archaeological and historical structures, such as churches, temples, and wells built in St. Paul's honor, can be found in the countries where he traveled. Young people can benefit from cultural and educational exchange programs. The involvement of 11 countries from two continents in this project will result in a serious cultural and educational exchange for all tourist groups. A single journey will allow them to experience European and Asian cultures, cultural transitions, and interactions. It is likely to be preferred in terms of tourism because it will bring together two different continents and various nations that have been in constant interaction throughout history. The route's cultural diversity and richness are at a level that will appeal to people of all ages and faiths. It is likely to be a model and innovative project because it will encourage historically and politically contentious countries to collaborate and contribute to peace.

We mentioned in the UNESCO and WHC criteria that it should have at least one item of outstanding universal value and that religious routes should include items ii, iv, and vi. St. Paul's Route differs from others in that it depicts the transformation of the most basic church architecture born in the Asian continent, in the Christian religion, into cathedrals in Europe, as well as a spiritual journey, as described in items ii and iv. It will also be the only route that takes tourists and pilgrims to Jerusalem, the holy city of Judaism, Christianity, and Islam. Another important aspect to which the route refers to item vi is that it involves conveying the beliefs, ideas, and arts of these three religions through an immersive and tactile experience. There is no need to be concerned about the route's authenticity, because St. Paul's journeys are described in detail in the Acts of the Apostles; if the route includes all of the stops described in the Bible, the information will be correctly conveyed to the participants. It is also possible to provide the integrity value: when the real stops described in the Bible, the monuments built in St. Paul's name, the areas where St. Paul preached, prayers, meditations, hymns, and stories are correctly transferred to the route, the route's integrity will be ensured. Meeting the conservation and management requirements may be the most difficult aspect of this route. The journey through 11 culturally and geographically diverse countries necessitates an excellent conservation and management strategy. The most important point is that all countries work together. Each country should establish an official institution to manage this route. This institution determines the route of the path in its own country, religious buildings to be visited, organizations such as ceremonies and celebrations, public order, location and quality of tourism facilities such as accommodation and food, the creation of buffer zones, infrastructure and transportation systems, tourist satisfaction, employment, and local people protection. Furthermore, it should assess the repair and restoration needs of the religious and architectural heritage along the route and prepare these structures for the route in a way that balances protection and use. In terms of the concept criterion, religious paths have their own core concepts. St. Paul's Route can promote international exchange and dialogue while describing the spread of Christianity that emerged in Asia to Europe.

The St. Paul's Route's distinguishing features are also very strong. The route's length and cultural diversity define the spatial character of the religious and cultural bond that remains between the two continents. The journeys made by St. Paul between 46 and 69 BC, as well as those made in his name today, show that there is enough existence for a historical identity. Its role and purpose are universal. It is at a level to demonstrate the products of Christianity in each country, as well as its interaction with societies and other monotheistic religions. The route's story will be shaped by stories about St. Paul in the Bible and among individuals. Scientific studies should be used to determine the route's boundaries and concentration points. The route should be accurate because it will convey information about a real person and events. For example, while St. Paul's Route forms a ring, travelers should be aware that the journey begins in the Tarsus District of Turkey, where St. Paul was born, and ends in Rome, where St. Paul was executed.

The ICOMOS and CIIC criteria emphasize authenticity as well. The use of the authentic environment is added to the authentic route. The authentic locations of St. Paul's Route

can also be determined using scientific research and information gathered from the local population, because he is a well-known character to almost everyone. If scientific sources can be used to determine how long St. Paul stayed in which place and what he did, how long the route will take and what can be done at each stop can be determined more clearly. Another requirement stated that continents should be the result of interactions between countries and regions. When the route is implemented in its authentic form, this feature will directly demonstrate how strong the route is as a route that spans two continents and twelve countries. ICOMOS's fourth criterion is that "affected cultures must maintain their tangible and intangible heritage values". This criterion was one of the strengths of the authentic St. Paul's Route, but it has been overlooked by the current St. Paul's Routes. First and foremost, the churches bearing the name of St. Paul should be included in the authentic route. St. Paul's Church and St. Paul's Well in Tarsus Town, Turkey, where he was born, and St. Paul's Church in Antiochia Ancient City of Isparta City, where he gave his first sermon, should be included on the route (Figure 6).

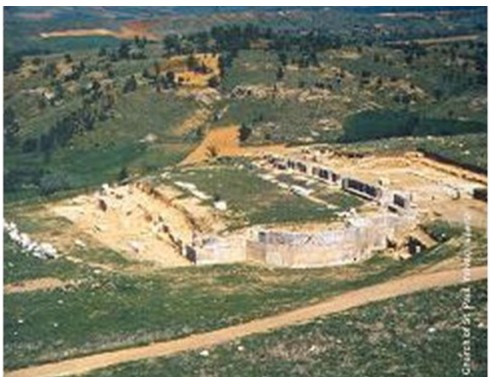 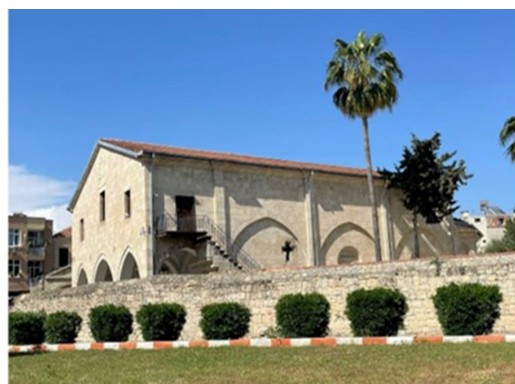

**Figure 6.** St. Paul Church—Isparta, Turkey; and St. Paul Church—Tarsus, Turkey.

One critical criterion is that the route generates revenue for the preservation and restoration of architectural heritage. Because the European continent is predominantly Christian, the church structures on this continent have been well preserved. However, because Islam is the dominant religion in Asian countries, many churches in these countries are neglected. Churches in various properties and borders are unable to be repaired due to political obstacles. As a consequence, St. Paul's Route could be a pioneer in the restoration of these churches with international cooperation. In Turkey, for example, there are numerous church structures dating from the ancient period to the end of the nineteenth century (Figure 7). Factors such as the deterioration of the Ottoman Empire's cosmopolitan structure at the end of World War I, the fact that the war environment and turmoil could not be overcome for a long time, and the migration of some peoples from different cultures left the churches in this geography without community, dysfunctional, and neglected. The Greek churches, in particular, were on the verge of collapse due to political, economic, and cultural issues between Greece and Turkey. İç Kale Church, Arap Evliyası Chapel, Civarda Burnu Chapel in Alanya and the Aya Georgios, Aya Yorgi, and Aya Baniya Churches in Isparta are among the religious structures in this situation. These churches were not built during St. Paul's lifetime, but they are included in the route as a result of St. Paul's efforts to spread Christianity.

It is critical to preserve the values of religious architectural heritage and intangible cultural heritage. The story of St. Paul's journeys is the route's most important intangible cultural heritage. Other elements should be researched scientifically and added to the route. ICOMOS and CIIC both value engagement with the natural environment. Other interesting natural areas discovered on the authentic route should be included in the new route. This criterion is easily met by the St. Paul's Route. In Turkey, for example, St. Paul's Route begins in Antalya's Perge Ancient City and ends in Isparta's Antiochia Ancient City. The natural areas where St. Paul actually walked are along the 500 km road between these

two stops. Travelers connect with nature and soil by camping in natural areas between these two stops. When the entire authentic route is researched, it will be possible to establish relationships with the natural and historical environments along the entire route. The St. Paul's Route already includes the criterion of encouraging interaction between people of different cultures and ethnic groups. With cultural interaction, Christianity, which originated in Asia, spread through the European continent. Except for the countries along the route, religious practices and traditions have spread across the entire European continent and other geographies since Christ. For centuries, both continents, and even the entire world, have seen Christian practices, rituals, and celebrations that are at the heart of the route.

| Name of Church | Construction Year | Plan | Photo |
|---|---|---|---|
| Alanya İç Kale Church | 11th Century | | |
| Alanya Arap Eviyası Chapel | 11th Century | | |
| Alanya Cilvarda Burnu Chapel | 13th Century | | |
| Isparta Aya Georgious Church | 1805 | | |
| Isparta Aya Yorgi Church | 1860 | | |
| Isparta Aya Baniya Church | 1865 | | |

**Figure 7.** Examples of churches that need restoration In Turkey.

Consequently, the St. Paul Route, like the Santiago de Compostela Pilgrim Route, has the potential to receive certificates from three international decision-making organizations: COE, UNESCO and WHC, and ICOMOS and CIIC. When combined with its authentic route for receiving certificates, it could be the world's second route to achieve three certificates. It can make significant contributions to the preservation and survival of architectural, religious, natural, and intangible cultural heritage across a broad geographic area, as well as to the establishment of world peace, international relations, and inter-religious dialogue.

**Funding:** This research received no external funding.

**Institutional Review Board Statement:** Not applicable.

**Informed Consent Statement:** Not applicable.

**Data Availability Statement:** Not applicable.

**Conflicts of Interest:** The authors declare no conflict of interest.

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
