# Peer review of "Determining the Characteristics of Faith-Themed Routes in Order to Receive an International Certificate: Studies on St. Paul’s Travels"

_religions, doi:10.3390/rel14091097_

Round 1

Reviewer 1 Report

- Sometimes the analysis tends to have a normative character that it is better to be avoided. Taking away such phrases will not devalue the article. See:

a. "Therefore, new religious paths should be (is possible to be)established..." (l. 12)

b. Tt will be an exemplary route to the entire world with these characteristics (l. 21)

etc...

- Editing of English language will clarify some lines, ideas, and concepts:  (see: "In addition to work on routes to document and register tangible and intangible religious values to ensure their involvement in sustainable tourism". [l. 38]

"Through sustainable tourism, these routes bring the tangible and intangible heritage of religion into the lives of modern people and continue the historical cycle."[l. 49]

and many European countries were built for his honor [l.57]

- "To truly understand the purpose, story, goals, 72 and consequences of St Paul's travels, they must be brought to life with all the stops. Because 73 of political and economic factors, St Paul's travels have not been addressed in a comprehen- 74 sive and realistic manner". = the sentence has to be restructured and rephrased  

-The St Paul's Route is a real historical route

-made four international trips [missionary journeys]

-judged by Roman soldiers (authorities) [l. 590]

-Although St Paul is a reasonable distance from the av- [l.684] erage, it should be designed so that travelers can join the route at any time.

- St Paul's Route has a strong enough concept to em- [l. 730] phasize exchange and dialogue between countries while bringing Christianity born in [731] Asia to Europe. = please rephrase.

- The section 4. Analysis of Certified Pilgrimage Routes provides general information (without bibliographical refs.). Τhis section should be drastically reduced.

-Please clarify the phrase: "Furthermore, it [l.708] will be the only route that takes tourists and pilgrims on the route to the origins of the [709] three great monotheistic religions." 

- which "great monotheistic religions", and which countries are the "origins"?

Author Response

Thank you very much for your refereeing. I tried to fulfill your criticisms as much as possible. I hope it turned out the way you wanted. Your criticisms and revisions are detailed in the attached file.

Reviewer 2 Report

The topic of the paper, St Paul`s Route, is interesting and important for European cultural heritage. Literature review and figures in the paper add value to the text.  In general , it is a good paper.

However, there are 2 things to be changed undoubtedly. Firstly,  The goal of the paper is presented at verse 636! Why so late? It should be in the first paragraph and in the abstract. Secondly,  Section 5.1.2 - It is a biography of St . Paul. Why here? I suggest to put it into introduction or into only one section. This paragraph (597-603) has information which is also in other parts of the paper . I suggest describing the figure of Paul only in one place, in a specified section. Finnally, I would also recommend to rethink verse 90. „Tragedy” it is a strong word. Are the authors sure?

Author Response

(The authors gave the same response as above.)
